# Age and Sex as Determinants of Acute Domoic Acid Toxicity in a Mouse Model

**DOI:** 10.3390/toxins15040259

**Published:** 2023-04-01

**Authors:** Alicia M. Hendrix, Kathi A. Lefebvre, Emily K. Bowers, Rudolph Stuppard, Thomas Burbacher, David J. Marcinek

**Affiliations:** 1Department of Environmental and Occupational Health Sciences, University of Washington, Seattle, WA 98195, USA; 2Environmental Fisheries Science Division, Northwest Fisheries Science Center, National Marine Fisheries Service, National Oceanic and Atmospheric Administration, Seattle, WA 98112, USA; 3Department of Radiology, University of Washington School of Medicine, Seattle, WA 98195, USA

**Keywords:** domoic acid, aging, seizures, excitotoxicity, amnesic shellfish poisoning

## Abstract

The excitatory neurotoxin domoic acid (DA) consistently contaminates food webs in coastal regions around the world. Acute exposure to the toxin causes Amnesic Shellfish Poisoning, a potentially lethal syndrome of gastrointestinal- and seizure-related outcomes. Both advanced age and male sex have been suggested to contribute to interindividual DA susceptibility. To test this, we administered DA doses between 0.5 and 2.5 mg/kg body weight to female and male C57Bl/6 mice at adult (7–9-month-old) and aged (25–28-month-old) life stages and observed seizure-related activity for 90 min, at which point we euthanized the mice and collected serum, cortical, and kidney samples. We observed severe clonic–tonic convulsions in some aged individuals, but not in younger adults. We also saw an association between advanced age and the incidence of a moderately severe seizure-related outcome, hindlimb tremors, and between advanced age and overall symptom severity and persistence. Surprisingly, we additionally report that female mice, particularly aged female mice, demonstrated more severe neurotoxic symptoms following acute exposure to DA than males. Both age and sex patterns were reflected in tissue DA concentrations as well: aged mice and females had generally higher concentrations of DA in their tissues at 90 min post-exposure. This study contributes to the body of work that can inform intelligent, evidence-based public health protections for communities threatened by more frequent and extensive DA-producing algal blooms.

## 1. Introduction

Harmful algal blooms (HABs) are phenomena in which marine or aquatic algal species rapidly produce large amounts of biomass and/or potent toxins that can adversely impact ecosystems, wildlife, and human health. One such HAB toxin is domoic acid (DA), an excitatory neurotoxin naturally produced by marine algae in the genera *Pseudo-nitzschia* and *Nitzschia* [1,2,3]. During DA contamination events, filter feeding organisms such as clams and planktivorous fish consume toxic *Pseudo-nitzschia* and can transfer DA to higher trophic levels including humans [4,5,6]. Acute DA poisoning in human seafood consumers is called Amnesic Shellfish Poisoning (ASP) and was first described following a human poisoning event in 1987 in Prince Edward Island (PEI), Canada. Clinical signs of ASP included gastrointestinal symptoms, such as vomiting, abdominal cramps and diarrhea, and neurologic symptoms, such as confusion, disorientation, loss of short-term memory, seizures, and coma [7].

In addition to identifying DA as a seafood toxin and characterizing its associated acute toxicosis, the canonical PEI poisoning event also offered some initial evidence of interindividual susceptibilities to the toxin. Both male sex and increased age (in 10-year increments) were positively correlated with memory loss (estimated odds ratio [OR] of 4.4 and 1.6, respectively) and hospitalization (estimated OR of 16.9 and 2.3, respectively) following toxin exposure [7]. At the time, researchers suggested that age-associated susceptibility might be mediated by age-associated renal function impairment, and that male reluctance to seek medical care might have biased data toward more severe symptoms in men [7]. Since then, follow-up studies on age- and sex-associated susceptibility to acute DA poisoning have been very limited [8].

Recent consumption surveys distributed by the Washington Department of Fish and Wildlife suggest that some commercial and recreational razor clam harvesters in the state are regularly exposed to DA at or above the current acute reference dose, i.e., at or above levels deemed safe for human consumption [9]. Further, the surveys reveal that consumers over the age of 60 are commonly represented in this exposed group [9]. Additional 2022 surveys in China estimated higher dietary DA exposures in consumers of more advanced age, though overall estimated DA intake was still quite low [10]. As human exposure to DA is occurring and will likely increase as oceans warm and HAB frequency accordingly increases, it is imperative that we clarify the factors contributing to interindividual DA susceptibilities. This will allow us to create the most appropriate protective and useful public health guidelines. These findings could also be important for marine mammal health as it is well known that California sea lions are frequently exposed to DA as well, and regularly experience DA toxicosis [11]. To address the knowledge gaps in possible age- and sex-associated susceptibilities to DA, we conducted a dose–response study with adult and aged male and female C57Bl/6 mice, assessing gross toxicity and seizure-related behavior.

## 2. Results

### 2.1. Associations of Dose, Age, and Sex with Clonic–Tonic Convulsion Risk

One hundred mice were monitored for 90 min following DA injection. If during that time a mouse experienced clonic–tonic convulsions (CTCs) lasting longer than 30 s, the mouse was euthanized in accordance with the protocol approved by our Institutional Animal Care and Use Committee. None of our adult mice of either sex experienced CTCs that mandated early euthanasia, but seven aged female mice and one aged male mouse did (Figure 1a,b). Convulsions were only observed at doses of 2.0 mg/kg body weight (bw) or greater, regardless of sex (Table 1; Figure 1a,b).

We used a survival analysis to examine the association of dose and sex with CTCs in more detail. We excluded age because CTCs were not observed in any adult mice. We modeled dose- and sex-associated CTC hazard ratios (HRs) using cox proportional hazard (CPH) regression. We first investigated a model with dose–sex interaction, but ultimately decided against it because of the small sample size and a concern for overfitting. We subsequently assessed the model with both dose and sex included as independent predicter variables; this model indicated a significant effect of dose, but not of sex (*p* = 0.0062 and *p* = 0.28, respectively; Appendix A), and the likelihood ratio test (LRT) did not indicate that including sex in the model significantly improved fit beyond that of the dose-only model (*p* = 0.24; Appendix A). This suggests that dose is the strongest predicter of CTC occurrence in aged mice, and we selected the dose-only model accordingly. This model estimates that a 1 mg/kg increase in DA dose is associated with a 16-fold increased risk for CTC occurrence (HR = 16.3 [2.9, 90.4], *p* = 0.0015; Appendix A).

Our inability to observe an association of sex with CTC occurrence was likely influenced by the fact that no male mice were exposed to the highest DA dose (2.5 mg/kg bw DA), a dose at which 75% of female mice showed CTCs (Figure 1). Overall CTC occurrence in males was therefore quite low.

### 2.2. Association of Dose, Age, and Sex with Hindlimb Tremor Risk

At every DA dose, aged mice exhibited hindlimb tremors (HLTs) at comparable or greater rates than their adult counterparts. In addition, female aged mice exhibited HLTs at comparable or greater rates than aged males (Figure 1c,d). The lowest observed adverse effect levels (LOAELs) for aged mice were lower than those for adults of the same sex, and LOAELs reported for adult females were lower than those for adult males (Table 1). Aged females and males both had LOAELs of 1 mg/kg bw (Table 1).

We used CPH regression methods of survival analysis to examine our HLT data. Unlike for CTCs, because both adult and aged animals exhibited HLTs, we were able to include dose, age, and sex as independent variables. Regression models with three- and two-way interactions failed to show significant interaction effects (Appendix A), and, as in the case of CTCs, raised concerns about overfitting due to small sample size. We therefore examined additive models, with dose, age, and sex as independent variables without interaction. Across all models explored, an independent effect of dose was consistently indicated (Appendix A). The fit of the full model including all three variables was significantly improved as compared to all nested models (see Appendix A for LRT *p*-values comparing models with and without variables in turn). The model including dose, age, and sex as independent variables with no interaction indicated that, by adjusting for each of the two remaining variables, a 1 mg/kg increase in DA dose was associated with an almost four-fold greater risk of HLTs (HR = 3.74 [2.00, 6.99], *p* = 3.6 × 10^−5^), aged animals had a three-fold greater risk of HLTs (HR = 3.07 [1.30, 7.24], *p* = 0.010), and females had an approximately 30% greater risk of HLTs (HR = 0.29 [0.11, 0.81], *p* = 0.019; Appendix A). Analysis of Schoenfeld residuals suggests that the sex variable, however, may violate the proportional hazard assumption of CPH models, meaning that, though no interactions between sex and other variables were indicated (Appendix A), the HRs reported here should be interpreted as weighted averages of the two sexes’ true hazard ratios over the observation period.

### 2.3. Association of Dose and Age with Mean and Maximum Seizure Scores

Over the course of 90 min following DA injection, we observed mice for 1-minute periods at each of the 1-, 5-, 10-, 30-, 40-, 50-, 60-, and 85-minute marks, scoring their seizure-related symptoms on a scale from 0 to 5 (see Section 4.6 for more details). Data from 86 mice were included in a subsequent analysis of the mean and maximum seizure scores recorded for each mouse during the periods.

Mean seizure scores, indicative of both the severity and persistence of symptoms experienced, ranged from 0 to about 2 in female mice (Figure 2a). Initial two-way Analysis of Variance (ANOVA) with dose, age, and the dose and age interaction as independent variables did not indicate a statistically significant interaction effect (F(1, 23) = 1.11, *p* = 0.30). We subsequently ran a two-way ANOVA with just dose and age as independent variables, and found a significant main effect of age (F(1, 24) = 24.49, *p* = 4.7 × 10^−5^) but not of dose (F(3, 24) = 0.87, *p* = 0.47). The results of post hoc pairwise t-tests indicated significantly higher mean seizure scores for the aged female mice compared to adult female mice dosed at 1.5 mg/kg bw DA (*p* = 0.00011; Figure 2a).

Trends were nearly identical for maximum seizure scores in female mice (Figure 2b). Two-way ANOVA with interaction again did not indicate a significant interaction effect (F(1, 23) = 0.33, *p* = 0.57). Subsequent two-way ANOVA without interaction again indicated an effect of age on maximum seizure score, but not of dose (F(1, 24) = 16.85, *p* = 0.00040 and F(3, 24) = 1.89, *p* = 0.16, respectively). Finally, as for mean seizure score, post hoc pairwise t-tests identified significantly greater maximum seizure scores for aged female mice compared to adult female mice administered 1.5 mg/kg bw DA (*p* = 0.0078; Figure 2b).

As for females, mean seizure scores for male mice ranged from 0 to about 2 (Figure 2a). Two-way ANOVA with interaction did not identify a significant effect of a dose and age interaction on mean seizure score (F(3, 31) = 0.079, *p* = 0.97). However, the main effects of both dose and age were identified in an additive ANOVA (F(3, 34) = 2.90, *p* = 0.049 and F(1, 34) = 6.77, *p* = 0.014, respectively). Pairwise t-tests did not, however, identify significant differences in direct comparisons.

Finally, two-way ANOVA with interaction did not identify a significant effect of a dose and age interaction for maximum seizure scores for male mice (F(3, 31) = 0.554, *p* = 0.65; Figure 2b). Two-way ANOVA without interaction did not show evidence for main effects of dose or age on male mice’s maximum seizure scores, either (F(3, 34) = 0.994, *p* = 0.41 and F(1, 34) = 3.56, *p* = 0.068, respectively).

### 2.4. Association of Dose and Age with Concentrations of DA in Tissues

We quantified DA concentrations (wet weight) in the serum, cortex, and kidneys of all mice at the time of their euthanasia. As expected, tissue DA concentrations in all 20 control animals administered saline were below detection limits. Data from 68 remaining animals that had been exposed to DA were analyzed for trends in tissue DA concentrations.

#### 2.4.1. Concentrations of DA in Female Mouse Tissues 90 Min Post Exposure

Domoic acid concentrations in the serum of female mice at 90 min post injection ranged from just over 400 ng/g to levels below detection (Figure 3). Our initial two-way ANOVA assessing the effect of dose, age, and the dose and age interaction on DA concentrations in the serum did not indicate a statistically significant interaction effect (F(1, 23) = 0.728, *p* = 0.40). Our additive ANOVA indicated a near significant effect of age on serum DA concentrations (F(1, 24) = 4.04, *p* = 0.056), and no effect of dose (F(3, 24) = 0.62, *p* = 0.55). Serum DA concentration was not associated with mean seizure score in either the adult or aged females (Appendix A).

Domoic acid concentrations in the right cortices of female mice euthanized 90 min after DA injection were all less than 30 ng/g, and 15 were below detection limits; these are noticeably lower than concentrations in serum (Figure 3). Initial two-way ANOVA assessing the effect of dose, age, and the dose and age interaction on DA concentrations in cortices did not indicate a statistically significant interaction effect (F(1, 24) = 2.83, *p* = 0.11). Additive ANOVA indicated a significant main effect of age (F(1, 25) = 21.26, *p* = 0.00011) but not of dose (F(3, 25) = 0.40, *p* = 0.76). Post hoc pairwise t-tests indicated significantly higher cortical DA concentrations for aged females compared to adult females at the 1.5 mg/kg bw dose (*p* = 0.0037; Figure 3).

Kidney tissue from female mice had the highest concentrations of DA (Figure 3). As for both serum and cortex, initial two-way ANOVA assessing the effect of dose, age, and the dose and age interaction on DA concentrations in kidney tissue did not indicate a statistically significant interaction effect (F(1, 24) = 0.63, *p* = 0.43). As for cortex, age was associated with kidney DA concentrations in the additive ANOVA (F(1, 25) = 10.86, *p* = 0.0029) but dose was not (F(3, 25) = 2.40, *p* = 0.092). Post hoc pairwise t-test showed significantly higher kidney DA concentrations in aged female mice compared to adult female mice at the 1.5 mg/kg bw dose (*p* = 0.018).

#### 2.4.2. Concentrations of DA in Male Mouse Tissues 90 Min Post Exposure

In general, the concentrations of DA measured in tissues from male mice were lower than those measured in tissues from female mice administered comparable doses (Figure 3). Age was also less clearly associated with tissue DA concentrations in male mice. For instance, DA concentrations in male mouse serum showed no significant association with dose, age, or a dose and age interaction (additive dose F(3, 34) = 0.47, *p* = 0.71; additive age F(1, 34) = 1.91, *p* = 0.18; interaction F(3, 31) = 0.96, *p* = 0.43). Additionally, all DA concentrations in male cortical samples were below detection limits. For male kidney DA concentrations, initial two-way ANOVA assessing the effect of dose, age, and the dose and age interaction did not indicate a statistically significant interaction effect F(3, 31) = 0.69, *p* = 0.57). For the additive model, neither dose nor age was found to be associated with DA concentrations (F(3, 34) = 2.50, *p* = 0.076 and F(1, 34) = 0.034, *p* = 0.85 for dose and age, respectively). Unlike females, serum DA concentrations did correlate with mean seizure scores in adult male mice, though not in aged males (Appendix A).

#### 2.4.3. Concentrations of DA in the Cortices of Mice Euthanized Early for CTCs

Domoic acid concentrations were also quantified in the six cortical samples from mice euthanized early for CTCs that contained enough material for analysis. As expected, these values tended to be higher than those observed in tissues collected 90 min after DA exposure from mice that did not have to be euthanized early. Cortical DA concentrations in aged female mice euthanized early for CTCs were 41.5 and 28.0 ng/g in two of the mice administered 2.0 mg/kg bw DA, and 64.9, 36.7, and 37.4 ng/g in the three mice administered 2.5 mg/kg bw DA. The DA concentration in the cortex of the one aged male mouse that was euthanized early for CTCs, which had been administered a DA dose of 2.0 mg/kg bw, was 16.9 ng/g.

## 3. Discussion

This is the first laboratory study to directly assess age-associated DA toxicity in both female and male mice. We tested gross toxicity and seizure-related symptoms and found consistent evidence that aged mice are more susceptible to acute DA neurotoxicity, and that female mice are more sensitive than males. We also report evidence that aged animals maintain higher concentrations of DA in serum, cortical, and kidney tissue than their adult counterparts. These findings will help to identify risk factors for DA toxicity that should be considered when protecting communities exposed to DA, moving public health research forward in the face of increased HAB threats [12,13,14].

### 3.1. Dose-Dependent Acute Domoic Acid Neurotoxicity

Evidence for an association between DA dose and neurotoxic endpoints was most clear when analyses could include data from all mice, given all doses. Survival analysis, the most data-inclusive of our techniques, indicated an association between higher DA doses and a greater occurrence of both severe and moderate neurotoxic endpoints (CTCs and HLTs, respectively). Furthermore, while we saw a significant effect of greater DA dose on higher mean seizure scores in male mice (for whom all dose groups were analyzed), neither mean nor maximum seizure score was seen to be associated with DA dose in females, for whom analysis did not include aged females in the 2.0 and 2.5 mg/kg bw DA dose groups.

Our data also show surprisingly limited evidence for an association between DA dosing and concentrations of DA persisting in mouse tissues at 90 min post exposure; age proved a far stronger predictor of DA concentrations. In the case of females, this may again be due to the need to limit the range of DA doses included in the analysis, and to the high variability of DA concentrations in each tissue type for aged animals. In the case of males, this likely results from the fact that tissue DA concentrations were low across the board.

Additionally, the results reported here indicate that future research assessing low-level, subconvulsive DA exposures in aged mice should consider using doses lower than 1.0 mg/kg bw. Though studies in adult mice might suggest that doses over 1.0 mg/kg bw DA are appropriate subconvulsive exposures, these doses can cause moderate neurotoxic symptoms in mice of more advanced age (40% and 20% HLTs in aged female and male mice, respectively). Additionally, doses of 2.0 mg/kg bw DA and above should be especially avoided, as they elicit potentially lethal CTC outcomes (80% and 20% in aged female and male mice, respectively).

### 3.2. Age-Associated Susceptibility to Acute Domoic Acid Toxicity

Our reports of more severe and persistent neurotoxic symptoms following acute DA exposure in aged animals (as compared to adults) and our observations of higher DA concentrations in tissues from aged mice (at least in the case of our females) are consistent with clinical reports from the 1987 human DA poisoning event, and with the two laboratory studies that have assessed acute DA symptomology in aged animals since then. When age-associated DA susceptibility was first reported in humans in 1987, it was suggested that impaired renal function might mediate the relationship [7], because younger patients that experienced severe symptoms all had pre-existing illnesses that involved impaired kidney performance [7]. Later, in 2002, it was reported that hippocampal slices (CA1 region) from young (3 mo) Sprague Dawley rats had some capacity to attenuate DA toxicity with repeated exposures, while slices from aged rats (26–29 mo) did not [15]. Later, in 2007, Hesp et al. followed up on this work and administered DA intraperitoneally (IP) and intrahippocampally (IH) to young (2–3 mo) and aged (22–27 mo) male Sprague Dawley rats [16]. Aged rats demonstrated increased seizure activity and mortality following the IP injections but not the IH administration, suggesting to the researchers that age-associated susceptibility was likely due to reduced toxin clearance, not increased neuronal sensitivity [16]. Thus, the age-associated susceptibility reported here complements the prevailing hypothesis that slower elimination in older animals contributes to their more severe symptoms following acute DA exposure.

### 3.3. Sex-Associated Susceptibility to Acute Domoic Acid Toxicity

Our results point to possible greater DA susceptibility in female mice than in males and suggest that this may be particularly true when comparing mice of advanced age. These results are surprising, given that, though limited, earlier work assessing sex differences in DA susceptibility has generally suggested that males are more sensitive to the toxin [7,17,18]. While our direct comparisons between sexes were constrained by the fact that we tested females and males asynchronously and administered them slightly different doses, our data and previous studies suggest that the possibility of sex differences changing over the lifespan should be studied further.

In the PEI poisoning incident, men were more likely than women to experience hospitalization and memory loss. At the time, it was suggested that this might result from behavioral differences between sexes, as opposed to inherent differences in biological susceptibility; researchers speculated that men perhaps only reported illness or sought clinical care when their symptoms were more severe, as compared to women [7]. Then, in a 1991 study concerning psychoneuroendocrine regulation by the lateral septal area (LSA) of the brain, DA was used to induce lesions in female and male Sprague Dawley rats’ LSA regions. In addition to their primary outcomes regarding humoral immune responses, the researchers made the unexpected finding that male rats experienced greater cell loss in the LSA following DA infusion compared to females [17]. This work prompted a follow-up in a 2013 study—female and male Sprague Dawley rats were given low doses of DA (0, 1.0, 1.8 mg/kg bw, IP injection), and their behavior was observed for 3 h [18]. Although both sexes demonstrated an increase in locomotion, grooming activity, and stereotypic behavior, females demonstrated effects earlier than males [18]. However, overt toxicity (mortality) was more common in the male rats than the female rats [18].

There are multiple, not mutually exclusive, possible explanations for differences in female and male susceptibility to DA poisoning. First, as we discussed for aging, Baron et al.’s report of earlier symptom onset in females may imply meaningful differences in toxicokinetics and elimination pathways between the sexes [18]. The higher DA concentrations that we observed in tissues from female mice support this conclusion. Future work should consider impairment of DA clearance that develops in each sex over the lifespan to better explore this possibility.

However, factors other than toxicokinetics may play a role in sex differences in DA toxicity and may underlie, specifically, the development of sex differences with advancing age. Endogenous gonadal hormones have been demonstrated to protect rodent brains from excitotoxic injury. In 1999, it was reported that systemic kainic acid (KA) administration induced hilar dentate neuronal damage in castrated males and ovarectomized females. It was additionally shown that impacts on neuronal integrity in intact females varied depending on the point in the estrous cycle at which KA exposure occurred, and that injection of estradiol, progesterone, and estrogen with KA mitigated associated injury at some points in the estrous cycle [19]. Our observation of severe DA neurotoxicity in aged females may therefore be mediated by changes in sex hormones over the lifespan: our aged mice were 25–28 mo and females were likely experiencing advanced-age-associated hormone fluctuations [13]. Conversely, the female rats included in previous work, which indicated greater susceptibility in males, were significantly younger (14–18 weeks old). Females in earlier work were young enough that they may have experienced protection from higher levels of sex hormones [18]. Thus, it is possible that previous reports of male susceptibility are accurate at young adult and adult life stages, but that hormonal changes at more advanced ages change sex-associated susceptibility. This idea could be investigated further by including more granular age groups in future studies and covering more of the lifespan.

### 3.4. Conclusions and Implications for Human Health

Consistent with previous work, our study strongly suggests that advanced age is associated with greater neurotoxicity following acute exposure to DA. It also suggests that aging-associated impairment of renal function or elimination processes may be partially responsible for this. Unlike the admittedly limited previous literature though, our study also indicates that female mice are more sensitive to DA neurotoxicity than males. Because this sex difference was observed specifically in our aged animals, we suggest that it may be related to aged females’ reduced protection from sex hormones. Sex differences in DA susceptibility would then present differently as hormones fluctuate over the lifespan, and may depend on the age at which sexes are compared.

As ocean conditions continue to change, DA contamination of marine ecosystems is likely to increase as warmer ocean temperatures have been linked to geographically larger and longer lasting *Pseudo-nitzschia* blooms [20]. It is essential that we understand which groups within our communities will be most at risk from rising threats of increased toxin exposure through seafood so that we can take appropriate protective measures. This study highlights the need to consider multiple life stages, and specifically advanced life stages, as we design such measures. It also shows that sex differences may be more dynamic than previously presumed. Additional research to inform public health actions should include study designs that compare sexes at multiple points in the lifespan.

## 4. Materials and Methods

### 4.1. Study Design

Dose–response studies were conducted with 25 adult and 25 aged female mice administered 0–2.5 mg/kg bw DA between June 2019 and January 2020, and with 25 adult and 25 aged male mice administered 0–2.0 mg/kg bw DA between May and October 2020. On each experimental day, one to six mice underwent testing illustrated in Figure 4. In brief, each mouse was taken into a testing room, acclimated to test housing, IP injected with saline or DA, and then returned to their test housing and observed both in real time and via a video recording for 90 min. If CTCs associated with severe DA toxicity occurred and sustained for at least 30 s, animals had to be humanely euthanized prior to 90 min. Otherwise, all animals were euthanized and dissected at the end of the 90 min period. All animal handling and experimental procedures were performed in accordance with protocols approved by the Institutional Animal Care and Use Committee at our institution.

### 4.2. Test Animal Care

Adult (7–9-month-old) female and male and aged (25–28-month-old) female and male C57Bl/6 NIA mice were obtained from the National Institutes of Aging colony and housed in the controlled environment of our institution’s animal research facility. Mice were allowed to acclimate to this research facility for at least one week prior to use in dose–response experiments. Upon arrival at the facility and until they were used, mice were housed in groups as large as five, provided free access to a standard rodent diet (PicoLab^®^ Rodent Diet 20, LabDiet, Richmond, IN, USA) and water ad libitum, and were on a 12 h light/dark cycle. Thirty minutes prior to their injection, animals were separated into individual test housing so that they could acclimate. Test housing was identical to facility housing but lacked food and water access as consumption of either would influence toxin excretion rates.

### 4.3. Domoic Acid Dosing

Due to personnel limitations, we were unable to run both sexes simultaneously. Females received doses of 0, 1.0, 1.5, 2.0, and 2.5 mg/kg bw DA. These doses were selected since they were (1) expected to elicit a spectrum of both subconvulsive and convulsive endpoints, based on previous literature and our lab’s pilot work ([21,22,23] and recently reviewed in [8]), and (2) aligned with the doses used in a previous study in rats [16]. However, because severe CTCs requiring euthanasia occurred in several of our aged female mice given 2.0 and 2.5 mg/kg bw DA, our sample sizes in those two groups were reduced. Therefore, we chose to shift the range of doses administered to male mice downward to 0, 0.5, 1.0, 1.5, and 2.0 mg/kg bw DA.

Stock solutions of DA targeting 1 mg/mL were prepared by diluting > 90% pure powdered DA (Sigma Aldrich, St. Louis, MO, USA) in sterile water, and were then quantified by enzyme-linked immunosorbent assay (ELISA) in parallel triplicate dilution series. Dosing solutions of 0.125, 0.1875, 0.25, 0.3125, and 0.375 mg/mL were prepared by further diluting stock in sterile saline (0.9% sodium chloride; Hospira, Inc., Lake Forest, IL, USA). To allow for more accurate dosing, each mouse was weighed on the day of their injection.

### 4.4. Real-Time Observations and Video Recording

Mice were video-recorded for 90 min in their test housing immediately after DA injection. They were also observed in real time so that sustained CTCs greater than 30 s, which mandated early euthanasia to minimize suffering, could be identified. All mice were recorded in side view (Canon VIXIA HF G10, Canon U.S.A., Inc., Melville, NY, USA), and some mice were additionally recorded in overhead view (GoPro Hero, GoPro, San Mateo, CA, USA); there was no effect of camera on neurotoxic analyses. Though periodic video stoppage and battery replacement mandated some periods of interruption, the dual-video system provided a nearly continuous video recording. Manual recording of activity and seizure state was performed by researchers during any periods in which cameras failed. These periods were generally brief and occurred with comparable frequency between dose and age groups and are therefore unlikely to have impacted results.

### 4.5. Tissue Collection and Quantification of Domoic Acid

Mice were euthanized by cervical dislocation either at the end of the 90 min video recording period or following 30 s of sustained CTCs. Blood serum, brains (right and left cortices), and kidneys were collected, flash-frozen, and stored in a −80 °C freezer prior to DA extraction [24].

Domoic acid was extracted from serum, right cortices, and kidneys in the following manner: Serum was mixed 1:3 volume/volume with 50% MeOH and vortexed for one minute to form homogenate [25]. Cortices were pulverized on liquid nitrogen to form powder, then mixed with 50% MeOH in a 1:3 ratio by weight to form homogenate [25]. Kidneys were ground in a glass mortar and pestle with 50% MeOH in a 1:3 ratio to form a similar homogenate [25]. All homogenates were then spun at 12,000 rpm (max; 13,870 rcf) for 3 min in a desktop centrifuge (accuSpin Micro 17, Thermo Fisher Scientific Inc., Waltham, MA, USA) and the supernatant extract was stored in air-tight vials at −4 °C until ELISA quantification of DA.

Directly prior to DA quantification, supernatant extracts were removed from −4 °C storage and filtered through spin filters (Ultrafree-MC centrifugal filters, MilliporeSigma, Burlington, MA, USA) spun at 12,000 rpm (max; 13,870 rcf) for 3 min in a desktop centrifuge (accuSpin Micro 17, Thermo Fisher Scientific Inc., Waltham, MA, USA). Then, extracts were diluted 10-fold in dilution buffer [25]; this minimum dilution was chosen to eliminate matrix effects. Finally, concentrations of DA in ng/g were quantified in extracts using commercially available ELISA kits (Biosense^®^, Bergen, Norway and Abraxis^®^, Warminster, PA, USA) as per kit instructions. Detection limits for DA in sample material were 2 ng/g for Biosense^®^ kits and 6.8 ng/g for Abraxis^®^ kits; to ensure consistency, a 6.8 ng/g detection limit was enforced on all quantifications, regardless of kit.

### 4.6. Video Recordings Post Exposure

All videos collected during the post-exposure period were scored by a single researcher blinded to the animals’ treatment. Videos were reviewed for seizure-related activity (scored on a 0–5 modified Racine scale; Table 2) during 1 min periods at 1, 5, 10, 30, 40, 50, 60, and 85 min post-exposure [16,18,26]. For each 1 min period, the following outcomes were recorded: (1) presence of subconvulsive HLTs, (2) if present, the number of individual HLTs that occurred, and (3) the maximum seizure score that occurred within the 1 min period (Table 2).

### 4.7. Statistical Analysis of Behavioral Endpoints

#### 4.7.1. Statistical Analysis of Binary Behavioral Endpoints: Clonic–Tonic Convulsion and Hindlimb Tremor Occurrence

Clonic-tonic convulsion and HLT LOAELs were determined as the lowest DA dose after which at least one mouse in each age and sex group experienced a CTC or HLT, respectively.

Because mice were observed for different lengths of time, depending on whether they experienced CTCs, we conducted survival analysis on our CTC and HLT data using CPH regression [27]. For CTC analysis, observation of a CTC was recorded as an outcome event at the time it occurred, and no observation of a CTC was recorded as a censoring event (end of observation) at 90 min. For HLT analysis, observation of an HLT was recorded as an outcome event at the first 1 min observation period during which an HLT was seen, and no observation of an HLT was recorded as a censoring event at the end of the observation time (either the time of euthanasia, if a CTC but no HLT was observed, or 90 min).

We generated multivariable and single-variable CPH regression models with independent variables of interest (dose, age, sex, and possible interactions), and compared their fits with the LRT. Exponentiated coefficients from the CPH regressions are reported as CTC or HLT HRs associated with each variable. We reviewed Schoenfeld residuals and deviance residuals to evaluate proportional hazard assumptions and possible outliers for each selected CPH model. Because survival analysis struggles to appropriately model situations in which outcomes are rare or nonexistent, we could not estimate the age-associated HR for CTCs (no CTCs were observed in adult mice, see Section 2.1). All analyses were performed using the statistical program R (R Core Team 2021).

#### 4.7.2. Statistical Analysis of Continuous Behavioral Endpoints: Mean and Maximum Seizure Score

To capture information about the duration and severity of overall seizure-related responses that our mice experienced, we calculated mean and maximum seizure scores for each mouse across all of their eight 1 min observation periods (see Section 4.6). Because females and males were not run concurrently and were administered different ranges of doses (1.0–2.5 mg/kg bw DA for females, and 0.5–2.0 mg/kg bw DA for males), we analyzed results from females and males separately.

Two-way ANOVAs were performed to analyze the effects of dose and age with and without an interaction on mean and maximum seizure scores. We ran Shapiro–Wilk tests and reviewed QQ plots and residuals versus fit plots to evaluate ANOVA assumptions. Where ANOVAs indicated significant interaction effects or main effects of a variable, post hoc pairwise t-tests with Bonferroni correction for multiple comparisons were performed, assessing any differences between the adult and aged mice at each dose. All analyses were performed using the statistical program R (R Core Team 2021).

Because ANOVAs do not accommodate censoring, we could only include animals that had been observed during all eight 1 min observation periods in these analyses. This necessitated the exclusion of eight mice that were euthanized early and led to the exclusion of female groups administered 2 and 2.5 mg/kg bw DA from the analysis due to small sample sizes (<3). Finally, the video recording of one mouse in the adult female 1 mg/kg bw DA group experienced interruptions, and that mouse was excluded as well.

### 4.8. Statistical Analysis of Tissue Domoic Acid Concentrations

Our statistical analysis of DA concentrations persistent in tissues from adult and aged mice at 90 min post exposure was similar to that for mean and maximum seizure scores, described above. Females and males were analyzed separately. We performed two-way ANOVAs with dose and age as independent variables, with and without interaction. Assumptions were tested with Shapiro–Wilk tests and QQ plot and residuals versus fit plot assessments. Post hoc pairwise t-tests with Bonferroni correction were used to examine age-associated differences within each dose stratum. All analyses were conducted using the statistical program R (R Core Team 2021).

Tissue DA concentration analysis exclusions were the same as those for mean and maximum seizure score data (see Section 4.7.2), with the exception that the mouse whose video recording was interrupted could still be included in tissue analysis. Grubb’s test was used to identify potential outliers in our data from the remaining 69 animals. One outlying serum sample that did not contain enough material for accurate quantification was ultimately excluded from the analysis. Samples below the assay detection limit were assigned a value of 3.4 ng/g DA for the purposes of the analysis; this was half the detection limit of the assay (6.8 ng/g DA, see Section 4.5).

## Figures and Tables

**Figure 1 toxins-15-00259-f001:**
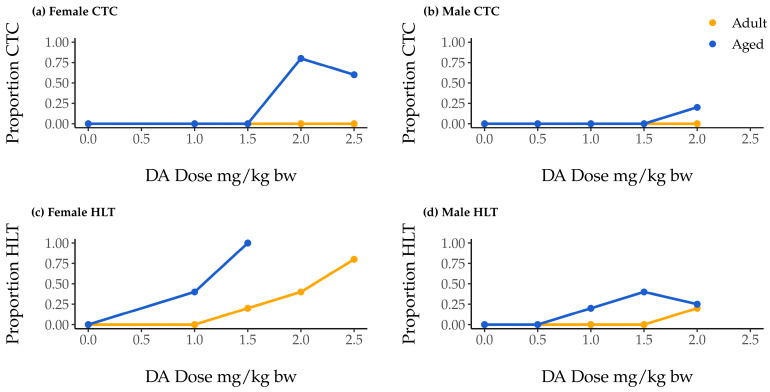
Proportion of aged (blue) and adult (yellow) female and male mice showing clonic–tonic convulsions (CTCs; (**a**) and (**b**), respectively) and hindlimb tremors (HLTs; (**c**) and (**d**), respectively) during the 90 min observation period following intraperitoneal (IP) injection with saline or domoic acid (DA; 1.0–2.5 mg/kg body weight [bw] for females, 0.5–2.0 mg/kg bw for males). Clonic-tonic convulsion data include all mice; HLT data are presented only for groups that maintained n = 3 after the exclusion of mice that could not be observed for a full 90 min (i.e., data are not presented for aged females given 2.0 and 2.5 mg/kg bw).

**Figure 2 toxins-15-00259-f002:**
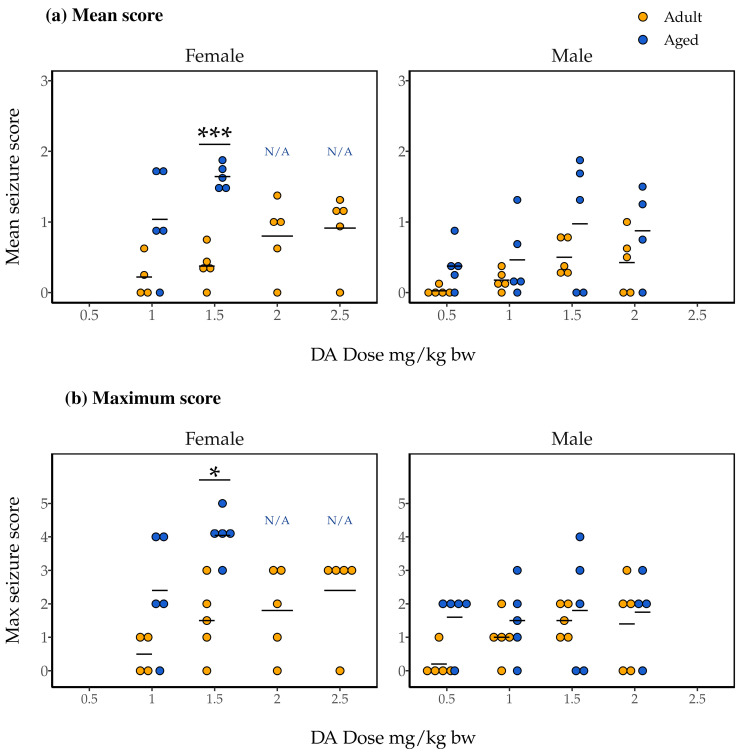
Mean (**a**) and maximum (**b**) seizure scores observed in aged (blue) and adult (yellow) female and male mice during 90 min of observation following IP injection with DA (1.0–2.5 mg/kg bw for females, 0.5–2.0 mg/kg bw for males). Data are presented only for groups that maintained n = 3 after the exclusion of mice that could not be observed for a full 90 min (i.e., data are not presented for aged females given 2.0 and 2.5 mg/kg bw; they are instead denoted not applicable (N/A)). Results of post hoc pairwise *t*-tests comparing adult and aged mice’s seizure scores within each sex and dose group are shown; * *p* < 0.05, *** *p* < 0.0005.

**Figure 3 toxins-15-00259-f003:**
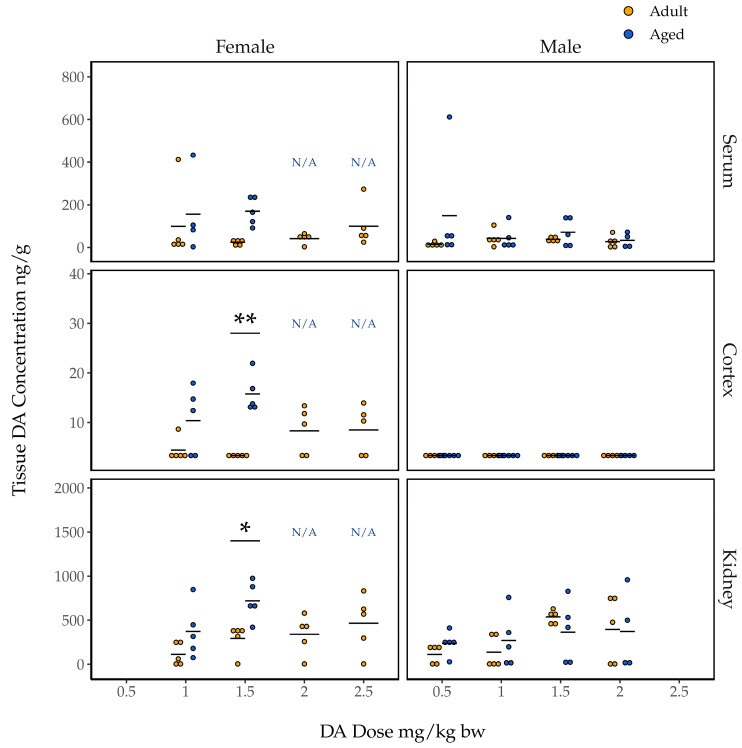
Domoic acid concentrations (n = 4–5; wet weight) quantified in the serum, right cortex, and kidneys of aged (blue) and adult (yellow) female and male mice 90 min after IP injection of DA (1.0–2.5 mg/kg bw for females, 0.5–2.0 mg/kg bw for males). Data are presented only for groups that maintained n = 3 after the exclusion of mice that were euthanized prior to 90 min post exposure due to sustained CTCs (i.e., data are not presented for aged females given 2.0 and 2.5 mg/kg bw; they are instead denoted N/A). One serum sample outlier from an aged female mouse in the 1.0 mg/kg group is omitted. Results of post hoc pairwise *t*-tests comparing adult and aged mice’s tissue DA concentrations within each sex and dose group are shown * *p* < 0.05, ** *p* < 0.005.

**Figure 4 toxins-15-00259-f004:**
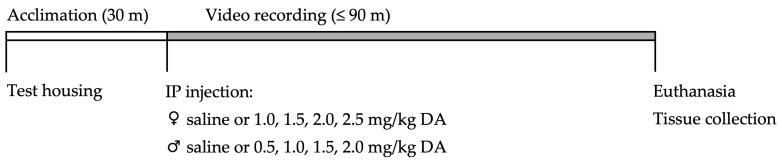
Day-of-experiment timeline of dosing and testing procedures for each mouse. Mice were brought into the experimental space, separated into and acclimated to test housing, administered saline or DA via IP injection, returned to their test housing, and observed both in real time and via video recordings for up to 90 min. If mice exhibited 30 s of sustained CTCs then they had to be humanely euthanized prior to 90 min. Otherwise, mice were euthanized at the end of 90 min. All animals were dissected, and serum, brain, and kidney samples were collected.

**Table 1 toxins-15-00259-t001:** Lowest observed adverse effect levels (LOAELs) associated with domoic acid (DA)-induced clonic–tonic convulsions (CTCs) and hindlimb tremors (HLTs) in each age and sex group. As no adult mice experienced CTCs, CTC-associated LOAELs are not applicable (N/A) to our adult mouse groups.

Group	CTC LOAEL	HLT LOAEL
Adult females	N/A	1.5 mg/kg DA
Aged females	2.0 mg/kg DA	1.0 mg/kg DA
Adult males	N/A	2.0 mg/kg DA
Aged males	2.0 mg/kg DA	1.0 mg/kg DA

**Table 2 toxins-15-00259-t002:** Modified Racine scale used to score symptom severity during 1 min observation periods at 1, 5, 10, 30, 40, 50, 60, and 85 min post DA exposure [16,23,26].

Score	State or Symptoms
0	No apparent effect
1	Pressed flat, little movement or stumbling walk
2	Hunched, head bobbing
3	Hindlimb tremors
4	Forelimb tremors and/or wet dog shakes
5	Clonic–tonic convulsions, rearing and falling, full-body shaking

## Data Availability

Data are available upon request to the corresponding author.

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
