# Peer review of "Age and Sex as Determinants of Acute Domoic Acid Toxicity in a Mouse Model"

_toxins, 2023, doi:10.3390/toxins15040259_

Round 1

Reviewer 1 Report

Review of toxins-2223287

Overall:
This study evaluates the effects of age, dose, and sex on seizure outcomes and tissue concentrations of domoic acid following IP injection. The authors measure seizure outcomes for 90 minutes following injection and record seizure severity until individuals are euthanized. Concentrations of DA are quantified in brain tissue, kidney, and serum.

The highest value finding of this paper is that CTCs were only observed in older female mice following high dose administrations. This is interesting and may as the authors point out be linked to hormone concentrations. It’s a shame the authors didn’t measure tissue concentrations of DA in those prematurely euthanized mice, while it would not be comparable to the other samples since the time after injection was different it would have potentially to be very informative looking at cortical concentrations in those individuals.

The other results the authors draw with hind limb seizures and tissue concentrations are less robust and convincing, not in part due to the fact that they don’t present or convey that statistical assumptions were checked for particular statistical tests.

I think the paper will be publishable after minor revisions to the statistics. My main recommendations are

-        Give detail to ensure statistical assumptions are met.

-        Give detail on how non-detects were treated mathematically (they should not be represented as 0, a correction factor should be applied.

-        A more thorough discussion on outliers, given that they represented 9% of observations. If they were suspected to be due to analytical errors, convince the readers that the other samples were handled appropriately.

Additionally, since the authors mention human exposure outcomes I would encourage them to examine literature on occurrence in marine mammals to see how their results could be informed by or inform studies in affected species such as California sea lions.

Specific comments

Line 30 – needs hyphen

Are HLTs observed in control mice for this line?

Table 1 and Table 2 could easily be combined

Figure 1 – in this reviewers opinion this should not be a line plot. There are no relationships between individuals in treatment groups so the lines connecting doses grouped by species draws a false idea of a continuous relationship or a temporal relationship between these variables. This figure would be better as a bar plot or dot plot (with error bars) like Figure 2.

Figure 2 – I don’t think you need the nd on this figure, it’s confusing especially since it appears to convey a seizure score value. The lack of points for females and the caption I feel adequately explain the absence of those data. Need to specify which post-hoc comparisons are being analyzed in figure caption – old vs young I’m assuming?

Figure 3 – same comment about nd as in Figure 2. Tissue concentrations are wet weight? Need to specify. It’s unclear how the authors did below detection limit corrections for figures. Should not be represented as 0.

Results –

Line 122 – This paragraph is all methodology

Line 133 – I’m not sure a paired t-test is being used appropriately here.

Line 171 – suggest concentrations rather than levels throughout

Line 184 – in a study examining interindividual variability in toxin response, why would you assume that differences in tissue concentrations are the result of human error and not real differences in toxin accumulation/depuration? If the authors have reason to suspect that nearly 10% of their quantitative data are suspect due to technical errors it does not inspire confidence about the rest of the data.

Line 188 - Our additive ANOVA did indicate a strong effect of age on serum DA levels (F(1,22)=14.74, p=0.00090).

Note that strong effect usually refers to the magnitude of the difference in the outcome variable, not the p or F value which is described as a significant effect. One can have a strong effect which is not significant and vice versa. Is the effect of age on serum DA concentrations strong, i.e. aged mice had 5x average concentrations in serum? I think it’s also worth noting here that this effect is only comparing two doses. The results here are also making me reevaluate Figure 3, the statistics reported indicated 22 DF but looking at Figure 3 it appears there are only 18 serum concentrations reported for female mice.

Line 247 – maybe this could be a figure? Relationship between serum DA concentrations and mean seizure score?

Line 294 – This sentence seems to be very speculative and doesn’t add a lot to the discussion.

Line 309 – do the authors have an explanation for sex based differences in toxicokinetics?

Line 323- mediated or exacerbated? Are the authors saying the symptoms were likely worse because of hormones? Or that the aged females were protected by hormones?

Line 343 - As ocean conditions continue to change…

This statement could use some sources and is also not well agreed upon by HAB biologists. Regional dependence in ubiquity, severity, and consistency is likely.

Methods 4.5 – are the authors following a standard method or established method here? Have the authors evaluated the stability of domoic acid in methanol?

https://www.ncbi.nlm.nih.gov/pmc/articles/PMC8399427/

https://www.sciencedirect.com/science/article/pii/S1568988302000306?casa_token=JQgZlsRBxwQAAAAA:GhAmHYS0qWuScgQTX9uxIgzGlj5LbdnRINcX7yW0ojslIYDPR3mDi1u648mAroelv1LxCrKMDQ

Methods 4.7.1

I would like to know that the authors checked assumptions of the Cox model here and that assumptions were met. Doesn’t have to be great detail, but as written it appears that they did not examine if assumptions were met.

Section 4.8

I would also like to know here that the authors checked assumptions of ANOVA models and ensured distributions of residuals et al were appropriate for post-hoc comparisons.

General Comments

This reviewer prefers nd = non detect, so for instances of no data where samples were not analyzed I suggest na or not analyzed. But again these abbreviations should not be used in the figures.

Reviewer 2 Report

Manuscript entitled „ Age and Sex as Determinants of Acute Domoic Acid Toxicity in a Mouse Model” is very interesting, well-written and well-planned experimental work. I fully support the publication of this manuscript; however I recommend the minor revision of manuscript. Small corrections should be made to the text according to the following comments:

Results

Line 66 – explain an abbreviation r IACUC

Table 1 – explain for the first time in the table an abbreviation DA

Line 92 – explain an abbreviation HR

Line 114 - explain an abbreviation LRT

Line 184, 200 – use an abbreviation DA instead of full name

Figure 1 – c) female HLT - why you did not show the results for aged group treated with 2.0 and 2.5 mg/kg bw of DA

Materials and Methods

Line 353 – write the total number of animals (male and female mice) which have been used in each experimental group studied

Line 399 – explain the full name of abbreviation ELISA

Line 426, 431 – replace comma in the numbers into a dot

Reviewer 3 Report

This is a very well-designed study with very interesting findings the effect of dose, age and sex as regards Domoic acid (DA) susceptibility, and the relevant implications for public health. There are only few points needing some improvement in order to make the manuscript more reader friendly.

Specific remarks:

1. Introduction

- Page 1, line 30: Not only the genus Pseudo-nitzschia spp. is responsible for DA production; other genera (e.g. Nitzschia spp.) are also DA-producers (see indicative ref. https://doi.org/10.1016/j.hal.2018.06.001). Please revise accordingly.

- Page 1, line 32: please place “Pseudo-nitzschia” in italics.

2. Results

- Page 2, line 66: please provide the exact reference for the IACUC guidelines.

3. Discussion

- Page 10, lines 308-309: Please provide the reference for Baron et al. by its number, according to the journal instructions.

4. Materials and Methods

- Page 11, lines 362-364: Please provide the experimentation license number (unless this information will be provided at the end of the manuscript, under ‘Institutional Review Board Statement’, which is not visible to me in this review copy).

- Page 11, lines 392-393: Please provide the reference for Petroff et al. (2021) by its number, according to the journal instructions.
